# Rethinking Self-Supervised Visual Representation Learning in Pre-training for 3D Human Pose and Shape Estimation

**Hongsuk Choi**[*†1,2], **Hyeongjin Nam**[*2], **Taeryung Lee**[2], **Gyeongsik Moon**[3], **Kyoung Mu Lee**[2,4]
[1]Samsung Research America, New York
[2]Dept. of ECE & ASRI, Seoul National University, Korea
[3]Meta Reality Labs Research
[4]IPAI, Seoul National University, Korea

## Abstract

Recently, a few self-supervised representation learning (SSL) methods have outperformed the ImageNet classification pre-training for vision tasks such as object detection. However, its effects on 3D human body pose and shape estimation (3DHPSE) are open to question, whose target is fixed to a unique class, the human, and has an inherent task gap with SSL. We empirically study and analyze the effects of SSL and further compare it with other pre-training alternatives for 3DHPSE. The alternatives are 2D annotation-based pre-training and synthetic data pre-training, which share the motivation of SSL that aims to reduce the labeling cost. They have been widely utilized as a source of weak-supervision or fine-tuning, but have not been remarked as a pre-training source. SSL methods underperform the conventional ImageNet classification pre-training on multiple 3DHPSE benchmarks by 7.7% on average. In contrast, despite a much less amount of pre-training data, the 2D annotation-based pre-training improves accuracy on all benchmarks and shows faster convergence during fine-tuning. Our observations challenge the naive application of the current SSL pre-training to 3DHPSE and relight the value of other data types in the pre-training aspect.

## 1 Introduction

Transferring the knowledge contained in one task and dataset to solve other downstream tasks (*i.e.*, transfer learning) has proven very successful in a range of computer vision tasks (Girshick et al., 2014; Carreira & Zisserman, 2017; He et al., 2017). In practice, transfer learning is done by pre-training a backbone (He et al., 2016) on source data to learn better visual representations for the target task. The ImageNet classification has been the de facto pre-training paradigm in computer vision, and the 3D human body pose and shape estimation (3DHPSE) literature has followed this.

Recently, self-supervised representation learning (SSL) has gained popularity in the interest of reducing labeling costs (Chen et al., 2020a; Grill et al., 2020; He et al., 2020; Caron et al., 2020; Hénaff et al., 2021). SSL pre-trains a backbone using unlabeled arbitrary object images and fine-tunes the backbone on target tasks. MoCo (He et al., 2020) and DetCon (Hénaff et al., 2021) surpassed the ImageNet classification pre-training for downstream tasks like object detection and instance segmentation on arbitrary class objects. Motivated by them, PeCLR (Spurr et al., 2021) and HanCo (Zimmermann et al., 2021) targeted a human hand and pre-trained a backbone on hand data without 3D labels. They showed the accuracy improvement for 3D hand pose and shape estimation from the controlled setting (Zimmermann et al., 2019), compared with random initialization (no pre-training) and the ImageNet classification pre-training. While the results of PeCLR and HanCo are promising for 3DHPSE, they have limited practical lessons. For example, the amounts of labeled hand data, which is fine-tuning data, are significantly smaller (∼64K) than that of the commonly used labeled body data (∼480K). Also, the total training (pre-training&fine-tuning) time of the different

---

[*]equal contribution.
[†]Most work was done when Hongsuk Choi was at Seoul National University.

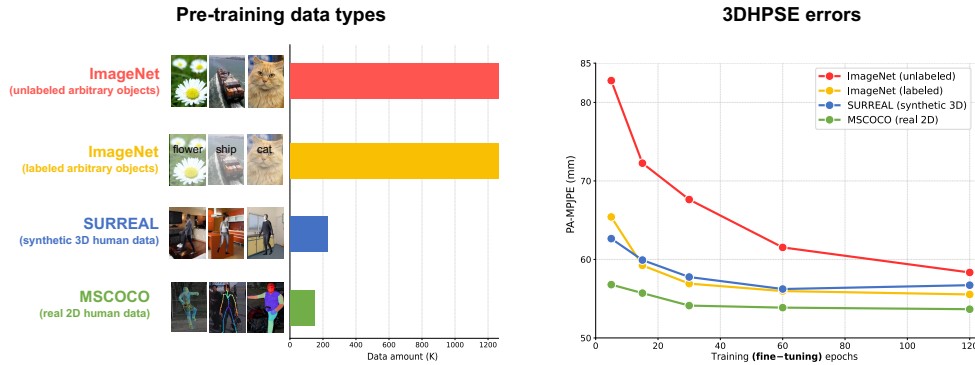

Figure 1: (Left) We pre-train a backbone (ResNet-50 (He et al., 2016)) with different data types: unlabeled arbitrary objects (Russakovsky et al., 2015), labeled arbitrary objects (Russakovsky et al., 2015), synthetic 3D human data (Varol et al., 2017), and real 2D human data (Lin et al., 2014). (Right) 3DHPSE errors when initializing its backbone with differently pre-trained weights. We fine-tune PARE (Kocabas et al., 2021) on Human3.6M (Ionescu et al., 2014) and MSCOCO (Lin et al., 2014) and evaluate it on 3DPW (von Marcard et al., 2018).

approaches is not matched, which is critical to the final accuracy (He et al., 2019). Last, they require labeled data with bounding boxes of a hand.

This paper questions the effectiveness of SSL pre-training for 3DHPSE by thoroughly comparing with alternatives in multiple aspects (*i.e.* final accuracy, convergence speed, and cost-effectiveness). We perform experiments by fixing the fine-tuning task to 3DHPSE and changing the pre-training approach. The experiments are organized in three steps. First, we compare state-of-the-art SSL methods, pre-trained on ImageNet, with the ImageNet classification pre-training. Different object detection and instance segmentation, the SSL methods are outperformed by the classification pre-training in three 3DHPSE benchmarks with 7.7% margin on average. Interestingly, the accuracy of SSL is comparable to or even worse than the random initialization baseline. The results imply general visual representations learned by SSL could be detrimental to 3DHPSE.

Second, we explore the reasons behind the current SSL methods' disappointing performance in depth by contriving a new pre-training approach. Modern SSL pre-training methods (Chen et al., 2020a; He et al., 2020) have two unfavorable factors on 3DHPSE; 1) they learn inconsistent representations for the same class instances as argued by (Khosla et al., 2020), which hinders learning high-level priors about a specific class, and 2) SSL pre-training has an instance-level learning characteristic (*i.e.*, a single attribute per image), which has an inherent task gap with 3DHPSE that requires understanding of the fine-level semantic information (i.e., multiple attributes per image), the human joints. We combine an SSL approach with 2D joint labels, which we call JointCon, to experimentally validate the two factors' effects. JointCon contrasts local image features of human joints instead of global image features of images.

Third, we compare SSL methods with 2D annotation-based pre-training and synthetic data pre-training on human data following PeCLR (Spurr et al., 2021) and HanCo (Zimmermann et al., 2021), and discuss cost-effectiveness in Section 5 and B. 2D annotation-based pre-training and synthetic data pre-training are worth investigating in that they share the motivation of SSL, which is to benefit from data[1] with less collection cost (Rong et al., 2019; Patel et al., 2021). In our experiments on human data, 2D annotation-based pre-training shows the highest accuracy and the fastest convergence speed among different pre-training approaches. Compared with the classification baseline, its final accuracy is increased by 3.1% on 3DPW (von Marcard et al., 2018) and 1.9% on Human3.6M (Ionescu et al., 2014). In 3DPW, the convergence speed is approximately 2× faster. In the semi-supervised setting, the accuracy improvement increases to 9.9% on 3DPW and 7.1% on Human3.6M. We assume rich pose and appearance information learned from the 2D pose data is the key to these improvements as expected. Synthetic data pre-training produces higher errors than the classification baseline. We conjecture that a domain gap between real and synthetic data inter-

---

[1] Unless otherwise noted, 'data' indicates labeled images

rupts efficient transfer learning. Finally, SSL on human data also underperforms the classification pre-training. The overall results of SSL suggest that the current stage of SSL may not be enough to benefit 3DHPSE, which essentially requires high-level understanding of human kinematic structure.

Our main empirical results are summarized in Figure 1. The current SSL that pre-trains on unlabeled arbitrary object images is not effective for 3DHPSE. Despite the least amount of pre-training data, 2D annotation-based pre-training provides the best result. This paper has two significant empirical contributions; 1) we provide novel experimental evidence and discussion points for people to rethink the naive application of SSL pre-training paradigm in 3DHPSE, and 2) we relight the value of other data types that have received relatively less attention in the pre-training aspect.

## 2 RELATED WORK

**3D human pose and shape estimation.** We focus on reviewing the body, not hand literature, where extensive works have been addressed for 3D pose and shape estimation from in-the-wild images. HMR (Kanazawa et al., 2018) proposed an end-to-end trainable human mesh recovery system that introduced adversarial loss to leverage MoCap data without images. GraphCMR (Kolotouros et al., 2019b) designed a graph convolutional network that takes the rest pose human mesh and image features as input and predicts mesh vertex coordinates. SPIN (Kolotouros et al., 2019a) combined a neural network regressor (Kanazawa et al., 2018) and an iterative fitting framework (Bogo et al., 2016). I2L-MeshNet (Moon & Lee, 2020) introduced a *lixel*-based 1D heatmap to estimate mesh vertex coordinates.Pose2Mesh (Choi et al., 2020) proposed a graph convolutional network that recovers 3D human pose and mesh from a 2D human pose. VIBE (Kocabas et al., 2020) and TCMR (Choi et al., 2021) extended HMR to video input. METRO (Lin et al., 2021) improved GraphCMR (Kolotouros et al., 2019b) with a transformer architecture. PARE (Kocabas et al., 2021) introduced a part-guided attention mechanism for mesh parameter regression. PyMAF (Zhang et al., 2021a) used mesh-aligned image features to iteratively refine prediction. 3DCrowdNet (Choi et al., 2022) resolved the inter-person occlusion issue in crowded scenes with 2D pose guidance for image features and a joint-based regressor (Moon et al., 2022a).

**Human dataset.** Human datasets for 3DHPSE can be broadly categorized into three types; Motion capture (MoCap) dataset, in-the-wild 2D dataset, and synthetic dataset. MoCap datasets (Ionescu et al., 2014; Mehta et al., 2017) provide accurate 3D joint labels with images captured from a controlled multi-view studio. In-the-wild 2D datasets (Andriluka et al., 2014; Johnson & Everingham, 2010) contain manually annotated 2D labels, which are usually 2D joints. MSCOCO (Lin et al., 2014) is the most widely used dataset that has rich 2D annotations, including part segmentation and DensePose (Güler et al., 2018). Synthetic datasets (Varol et al., 2017; Patel et al., 2021) render 3D human avatars on synthetic background images. Human poses from MoCap data and appearances from real scan data are exploited. Recent works (Patel et al., 2021; Baradel et al., 2021; Cai et al., 2021) have shown that fine-tuning on synthetic human data can improve accuracy on real-world benchmarks. 3DPW (von Marcard et al., 2018) is an in-the-wild human benchmark with 3D body pose and mesh annotations. Since few in-the-wild 3D human datasets exist, evaluation on 3DPW is the current best way to evaluate 3DHPSE methods on in-the-wild images.

The concurrent work of (Pang et al., 2022) provides experimental results on pre-training on three datasets (classification on ImageNet, 2D pose estimation on MPII (Andriluka et al., 2014) and MSCOCO). However, the effects of SSL and synthetic data are still unexplored. Considering that SSL has shown a powerful impact on other vision tasks, the extensive experiments and analysis on the current SSL methods distinguish our work from (Pang et al., 2022).

**Self-supervised representation learning.** Recently, contrastive learning-based methods are showing state-of-the-art performance among self-supervised approaches. The contrastive learning's fundamental idea is to pull together an anchor and a "positive" sample in embedding space, and to push apart the anchor from many "negative" samples (Khosla et al., 2020). Since this strategy can be applied to unlabeled training data, assuming a positive pair from data augmentations of the same sample, the research community has endeavored to use the learned representation for downstream transfer tasks. In practice, a backbone is pre-trained on a large-scale classification dataset (Russakovsky et al., 2015; Mahajan et al., 2018) without using labels, and fine-tuned for classification, object detection, or instance segmentation.

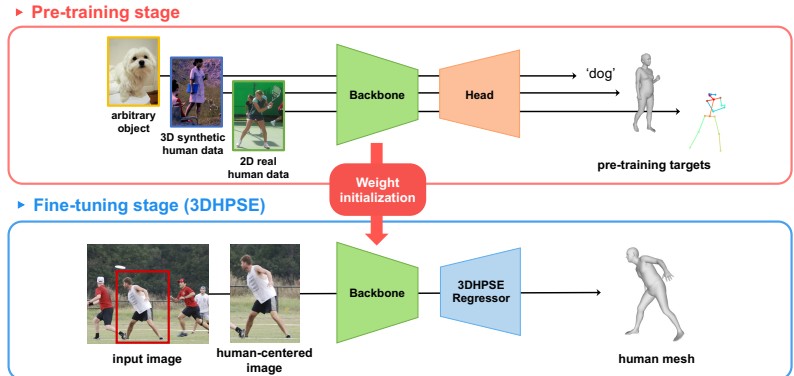

Figure 2: Overview of the training procedure for 3DHPSE. We pre-train a backbone with each different data type (*e.g.*, labeled arbitrary object, synthetic 3D human data, and real 2D human data). From the pre-trained backbone, we fine-tune both the backbone and a human mesh regressor in an end-to-end manner.

MoCo (He et al., 2020; Chen et al., 2020c) interpreted the contrastive learning as building a dynamic and large dictionary of embeddings with a queue and a moving-averaged key encoder. SimCLR (Chen et al., 2020a;b) introduced a nonlinear projection layer and proved that augmentation during pre-training should be stronger than supervised learning. SwAV (Caron et al., 2020), SiamSiam (Chen & He, 2020), and BYOL (Grill et al., 2020) eliminated the requirements for negative samples, while maintaining the Siamese architecture. DetCon (Hénaff et al., 2021) proposed a new contrastive objective that is based on unsupervised mask generation.

PeCLR (Spurr et al., 2021) and HanCo (Zimmermann et al., 2021) are the recent works that adapted SSL to 3D hand pose and shape estimation. PeCLR proposed a contrastive objective equivariant to geometric transformations (*e.g.*, rotation and translation), which models the transformations in a latent vector level similar to (Rhodin et al., 2018) and NSD (Rhodin et al., 2019). It showed accuracy improvements over the random initialization baseline, but trained on a small-scale dataset (∼32K) (Zimmermann et al., 2019) during fine-tuning, although more hand data, for example, MSCOCO (150K) and YT3D (47K), exists. HanCo applied MoCo (He et al., 2020) on a background-augmented unlabeled hand dataset, but the improvement was marginal and like PeCLR, experiments were done in the controlled setting, not the in-the-wild environment.

## 3 CONVENTION OF 3D HUMAN POSE AND SHAPE ESTIMATION

**Architecture.** A 3DHPSE network consists of a backbone and a human mesh regressor, as depicted in Figure 2. A backbone (He et al., 2016; Sun et al., 2019) extracts image features from a given human-centered image. The features are fed to a human mesh regressor that estimates a mesh defined by human models, such as SMPL (Loper et al., 2015).

**Pre-training.** Almost all 3DHPSE methods pre-train their backbones by ImageNet (Russakovsky et al., 2015) classification. They take pre-trained weights from the open source (*e.g.*, torchvision (Paszke et al., 2017)) and initialize their backbones with them in practice. Initializing a backbone's weights with the ImageNet classification-pre-trained weights is known to expedite training convergence and to bring better accuracy than random initialization.

**Fine-tuning.** During fine-tuning, a pre-trained backbone and a human mesh regressor are trained in an end-to-end manner by mix-using MoCap and in-the-wild 2D datasets. 2D annotations of in-the-wild 2D datasets weakly supervise the 3D mesh prediction (Kanazawa et al., 2018), and the 3D pseudo-mesh labels generated from 2D joints (Pavlakos et al., 2019; Joo et al., 2021; Moon et al., 2022b) directly supervise the 3D output. The mixed use of the MoCap and the in-the-wild 2D datasets has enabled reasonable performance on in-the-wild 3DHPSE, despite the scarce in-the-wild 3D training data.

# 4 EXPERIMENT

## 4.1 SETTING

We adopt ResNet-50 (He et al., 2016) as a backbone and a SMPL (Loper et al., 2015) mesh as an estimation target. During fine-tuning on 3DHPSE, we mainly use PARE (Kocabas et al., 2021) as a human mesh regressor and Human3.6M (Ionescu et al., 2014) and MSCOCO (Lin et al., 2014) as training datasets. For the semi-supervised setting, we exploit 10% of Human3.6M and 10% of MSCOCO data. 3DPW (von Marcard et al., 2018), Human36M (Ionescu et al., 2014), and MuPoTS-3D (Mehta et al., 2018) are used as evaluation benchmarks. We report PA-MPJPE for 3DPW, Human36M, and 3DPCK for MuPoTS-3D following the convention. PA-MPJPE stands for the Procrustes-aligned mean per-joint position error in millimeters. 3DPCK indicates the percentage of correct 3D keypoints.

**Pre-training.** For the SSL pre-training on ImageNet (Chen et al., 2020b; Chen & He, 2020; Caron et al., 2020; Chen et al., 2020c; Hénaff et al., 2021), we use the publicly released pre-trained ResNet-50 weights. SSL pre-training on ImageNet typically requires much larger computational costs to be effective than the classification baseline. For pre-training on human datasets, we fix the pre-training length (total number of training epochs) at 140 epochs regardless of methods, which is longer than those of 3DHPSE adapted SSL papers (100 epochs) (Spurr et al., 2021; Zimmermann et al., 2021) and the same as that of (Xiao et al., 2018). To pre-train the proposed 2D pose-driven alternatives, we decay the learning rate by $10\times$ at 90 and 120 epochs, starting from the initial value of $10^{-3}$ following (Xiao et al., 2018).

**Fine-tuning.** For fine-tuning, we train a 3DHPSE network for 120 epochs and decay the learning rate by $10\times$ at 90 epochs, which is originally $10^{-4}$. This schedule empirically showed full convergence for various networks initialized with different pre-trained weights. For the case of training from scratch (*i.e.*, random initialization), following the spirit of (He et al., 2019), we extend the total training length to 240 epochs and decay the learning rate by $10\times$ at 180 epochs from $10^{-4}$. It roughly matches the total training length of the pre-training counterparts (140 epochs + 120 epochs) and empirically showed full convergence.

## 4.2 PRE-TRAINING ON IMAGENET

We first examine the effects of the current state-of-the-art SSL methods (Chen et al., 2020b; Chen & He, 2020; Caron et al., 2020; Chen et al., 2020c; Hénaff et al., 2021) that pre-train a backbone on ImageNet. As summarized in Table 1 and Figure 3, when full fine-tuning data is used, they show approximately $2\times$ slower convergence and 7.7% higher errors (PA-MPJPE) than the ImageNet classification pre-training (He et al., 2016). Interestingly, the final accuracy is even worse than the random initialization baseline, though the convergence is faster. In the semi-supervised setting, SSL methods outperform the random initialization baseline by 2.5%, but still the classification pre-training provides the best overall accuracy.

We analyze the results that seemingly oppose the fact that SSL surpasses the random initialization and classification baselines in object detection and instance segmentation (Grill et al., 2020; Hénaff et al., 2021) by answering three questions. (i) ***Why SSL is worse than the random initialization baseline, when full fine-tuning data is used?*** We conjecture that a data domain gap is one of the reasons for the different results. While both object detection and instance segmentation target localization of arbitrary class objects, 3DHPSE only targets a single class, the human. For inference on arbitrary class objects, learning a wide range of general features unlimited to labels of a dataset could be advantageous in the generalization aspect (Tendle & Hasan, 2021). However, for 3DHPSE, a backbone network is preferred to learn more about human features rather than features of arbitrary objects, given the limited learning capacity. Transferring knowledge about arbitrary objects could distract a network from learning necessary human features for 3DHPSE. (ii) ***Why the classification pre-training outperforms the random initialization baseline, while SSL does not, when full fine-tuning data is used?*** The classification pre-training makes a backbone to learn high-level semantic representations, such as the global structure of objects, that could be beneficially transferred to inference on humans. For example, (Yosinski et al., 2015) showed that AlexNet (Krizhevsky et al., 2017) trained on ImageNet can recognize important features of human faces, although ImageNet has no labels of human faces. On the contrary, visual representations learned by SSL are likely to lack high-

Table 1: Effects of self-supervised pre-training on ImageNet. The red and blue colors indicate the first and second best scores, respectively.

| fine-tuning data | pre-training data | pre-training method | 3DPW PA-MPJPE↓ | H36M PA-MPJPE↓ | MuPoTS 3DPCK↑ |
|---|---|---|---|---|---|
| H36M+ MSCOCO (100%) | - | random init. | 56.37 | 52.72 | 67.12 |
| | ImageNet (labeled) | classification | 55.65 | 48.36 | 67.76 |
| | ImageNet (unlabeled) | SimCLR (Chen et al., 2020a;b) | 59.56 | 56.93 | 65.67 |
| | | SimSiam (Chen & He, 2020) | 56.42 | 51.82 | 66.34 |
| | | SwAV (Caron et al., 2020) | 56.85 | 52.18 | 66.19 |
| | | MoCo v2 (Chen et al., 2020c) | 58.34 | 55.43 | 66.68 |
| | | DetCon (Hénaff et al., 2021) | 64.54 | 58.83 | 63.83 |
| H36M+ MSCOCO (10%) | - | random init. | 73.37 | 67.59 | 57.32 |
| | ImageNet (labeled) | classification | 63.29 | 58.79 | 62.75 |
| | ImageNet (unlabeled) | SimCLR (Chen et al., 2020a;b) | 73.97 | 72.02 | 59.80 |
| | | SimSiam (Chen & He, 2020) | 66.94 | 62.45 | 63.34 |
| | | SwAV (Caron et al., 2020) | 68.96 | 63.26 | 60.79 |
| | | MoCo v2 (Chen et al., 2020c) | 64.76 | 60.70 | 63.47 |
| | | DetCon (Hénaff et al., 2021) | 81.63 | 82.60 | 55.03 |

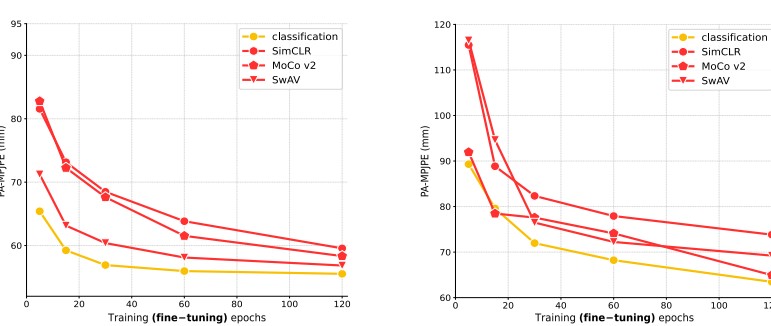

Figure 3: Learning curves of PA-MPJPE on 3DPW in the fine-tuning stage when using full fine-tuning data (Left) and 10% of fine-tuning data (Right). The backbone is initialized with different weights pre-trained on **ImageNet** by different pre-training methods.

level information that can be found in objects with the same class. Instead, SSL focuses on learning representations that are invariant to data augmentations, which are performed on a single object sample. These representations can be inconsistent over instances of the same class (Khosla et al., 2020). Considering that 3DHPSE essentially requires high-level priors that are shared across different human samples, inconsistent representations learned by SSL could be detrimental to 3DHPSE. (iii) *Why SSL is effective in the semi-supervised setting, compared with the random initialization?* Inconsistent representations learned by SSL could be a problem for 3DHPSE, but it does not indicate their representations are obsolete. A convolutional neural network requires sufficient training data to extract meaningful low-level features (*e.g.*, textures) from images. In this perspective, insufficient training images deprive a network of learning high-level representations, which in turn harms 3DH-PSE. In such a circumstance, SSL can step in to provide a sufficient amount of training images to a network and to make it learn necessary features.

## 4.3 ANALYSIS OF SSL FOR 3DHPSE.

We further investigate the reasons for the SSL's disappointing performance discussed in Section 4.2, with experimental evidence. We devise a new pre-training approach, JointCon, that combines the SSL approach with the 2D annotation-based approach. JointCon extracts joint-level features (Moon et al., 2022a) by sampling image features based on GT 2D joint locations. It applies contrastive learning in the joint-level features to pull together an anchor with "positive samples" and push

Table 2: Effects of different representation-level in pre-training via ablation study on JointCon. We use MSCOCO for pre-training. The red and blue colors indicate the first and second best scores, respectively.

| fine-tuning data | variations of JointCon | 3DPW PA-MPJPE↓ | H36M PA-MPJPE↓ | MuPoTS 3DPCK↑ |
|---|---|---|---|---|
| H36M+ MSCOCO (100%) | JointCon(I): instance-level | 58.29 | 53.17 | 67.06 |
| | JointCon(J): joint-level | 54.25 | 45.52 | 68.87 |
| | JointCon(I+J): instance-level + joint-level | 56.71 | 47.99 | 67.85 |
| H36M+ MSCOCO (10%) | JointCon(I): instance-level | 67.71 | 62.04 | 62.64 |
| | JointCon(J): joint-level | 57.19 | 54.97 | 67.98 |
| | JointCon(I+J): instance-level + joint-level | 59.59 | 56.55 | 67.38 |

apart the anchor from "negative samples". We design three variations of JointCon as below: 1) **JointCon(I)** defines "positive samples" as joint-level features extracted from the same image and "negative samples" as those from different images. 2) **JointCon(J)** defines "positive samples" as joint-level features extracted from the same joint label and "negative samples" as those from different labels. Note that it treats joint-level features of the same joint class from different images as "positive samples". 3) **JointCon(I+J)** defines "positive samples" as joint-level features extracted from the same image and the same joint label, and "negative samples" otherwise.

Table 2 shows that JointCon(J) outperforms the other variations in all three benchmarks. The accuracy gap between JointCon(J) and JointCon(I+J) proves that it is important to learn consistent representations across instances of the same class. Contrasting representations of the instances in the same class (*i.e.*, objects or joints in the same class) hinders a backbone from learning useful high-level priors and degrades the performance of 3DHPSE. As a result, the ImageNet classification pre-training rather learns consistent representations of the instances in the same class and achieves better accuracy than SSL, as shown in Table 1 The other important observation is that both JointCon(J) and JointCon(I+J) outperform JointCon(I). The accuracy boost by simply adding joint information to the SSL system demonstrates that the current SSL's low performance is due to the instance-level learning characteristic rather than network architecture, data augmentation, or losses.

## 4.4 PRE-TRAINING ON A HUMAN DATASET

We investigate three pre-training approaches, SSL, 2D annotation-based pre-training, and synthetic data pre-training. First of all, given the results of Section 4.2, we apply SSL on a human dataset following (Spurr et al., 2021; Zimmermann et al., 2021). PeCLR (Spurr et al., 2021) that adapted SimCLR (Chen et al., 2020a;b) to a 3D hand pose and shape estimation, MoCo v2 (Chen et al., 2020c), the method used by HanCo (Zimmermann et al., 2021), and Swav (Caron et al., 2020), which is not based on contrastive learning, are experimented. MSCOCO (Lin et al., 2014) is used as a pre-training dataset. For synthetic data pre-training, we pre-train PARE (Kocabas et al., 2021) on AGORA (Patel et al., 2021) and SURREAL (Varol et al., 2017). For 2D annotation-based pre-training, we experiment with part segmentation estimation, DensePose (Güler et al., 2018) estimation, and 2D pose estimation on MSCOCO.

As shown in Table 3 and Figure 4, the SSL methods bring faster convergence, but the final accuracy becomes on par with the random initialization baseline. It is also lower than the traditional classification pre-training. SSL appears to be effective only when significantly less fine-tuning data (10% Human3.6M + 10% MSCOCO, $< 50K$) is used. The results coincide with the experimental results of PeCLR and HanCo; PeCLR was fine-tuned on ~32K labeled data (Zimmermann et al., 2019), and HanCo was fine-tuned on ~64K labeled data (Zimmermann et al., 2021). Synthetic data pre-training shows a similar tendency with SSL, though it outperforms SSL. On the contrary, 2D annotation-based pre-training improves accuracy against the random initialization and classification baselines on all benchmarks. In the semi-supervised setting, the improvement increases up to 12.4% compared with the random initialization. The convergence speed is much faster than both random initialization and classification, especially in the semi-supervised setting.

Table 3: Effects of different pre-training schemes on a human dataset. The red and blue colors indicate the first and second best scores, respectively.

| fine-tuning data | pre-training data | pre-training method | 3DPW PA-MPJPE↓ | H36M PA-MPJPE↓ | MuPoTS 3DPCK↑ |
|---|---|---|---|---|---|
| H36M+ MSCOCO (100%) | - | random init. | 56.37 | 52.72 | 67.12 |
| | ImageNet (labeled) | classification | 55.65 | 48.36 | 67.76 |
| | MSCOCO (unlabeled) | PeCLR (Spurr et al., 2021) | 57.27 | 49.84 | 66.88 |
| | | MoCo v2 (Chen et al., 2020c) | 56.35 | 50.06 | 66.95 |
| | | SwAV (Caron et al., 2020) | 58.20 | 56.50 | 66.77 |
| | AGORA SURREAL | 3DHPSE | 56.77 | 51.96 | 67.80 |
| | | | 56.45 | 52.51 | 67.65 |
| | MSCOCO | part segmenetation est. | 54.37 | 48.25 | 68.55 |
| | | DensePose est. | 54.16 | 49.18 | 68.43 |
| | | 2D pose est. | 53.34 | 44.89 | 69.04 |
| H36M+ MSCOCO (10%) | - | random init. | 73.37 | 67.59 | 57.32 |
| | ImageNet (labeled) | classification | 63.29 | 58.79 | 62.75 |
| | MSCOCO (unlabeled) | PeCLR (Spurr et al., 2021) | 71.24 | 66.58 | 60.16 |
| | | MoCo v2 (Chen et al., 2020c) | 66.93 | 62.06 | 61.36 |
| | | SwAV (Caron et al., 2020) | 75.98 | 73.83 | 58.40 |
| | AGORA SURREAL | 3DHPSE | 64.71 | 63.17 | 65.73 |
| | | | 62.51 | 59.10 | 64.50 |
| | MSCOCO | part segmenetation est. | 56.80 | 54.10 | 67.69 |
| | | DensePose est. | 57.33 | 54.46 | 68.02 |
| | | 2D pose est. | 56.92 | 55.31 | 67.12 |

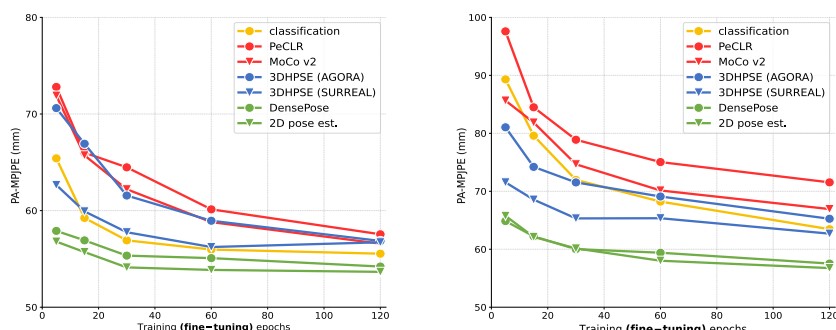

Figure 4: Learning curves of PA-MPJPE on 3DPW in the fine-tuning stage when using full fine-tuning data (Left) and 10% of fine-tuning data (Right). The backbone is initialized with different weights pre-trained on a **human dataset** by different pre-training methods.

We think the results of SSL on human data support the statement in Section 4.2 that representations learned by SSL are hard to embed high-level information related to humans. Synthetic data pre-training has potential to benefit from rich pose and appearance diversity, but a domain gap problem from highly different image appearances between synthetic and real images seems to remain. Compared with SSL and synthetic data pre-training, 2D annotation-based pre-training appears to effectively transfer high-level priors of humans, such as body articulation, to 3DHPSE. Well-aligned 3D body pose and robustness to occlusion in Figure 5 verify the effectiveness of 2D annotation-based pre-training. Figure 6 visualizes each human part's feature activation of PARE (Kocabas et al., 2021) to further inspect how representations learned by each pre-training approach affect the human mesh regressor's understanding of human geometry.

## 5 DISCUSSION

Our empirical observations propose that we should explore 2D annotation-based pre-training instead of SSL for 3DHPSE. However, SSL may still have an advantage over 2D annotation-based pre-

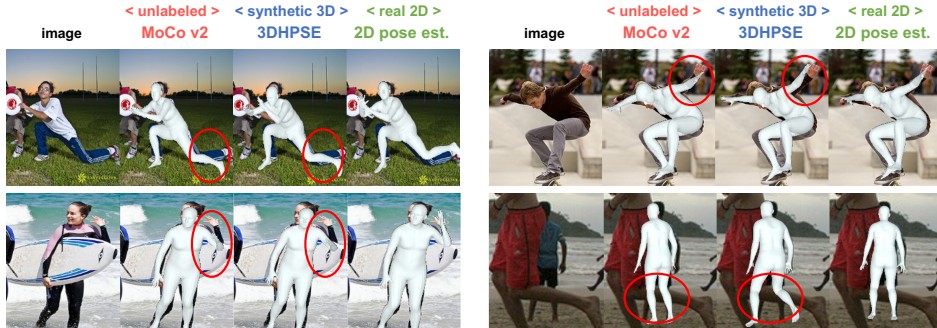

Figure 5: Qualitative comparison among different pre-training schemes. We highlighted their representative failure cases with red circles.

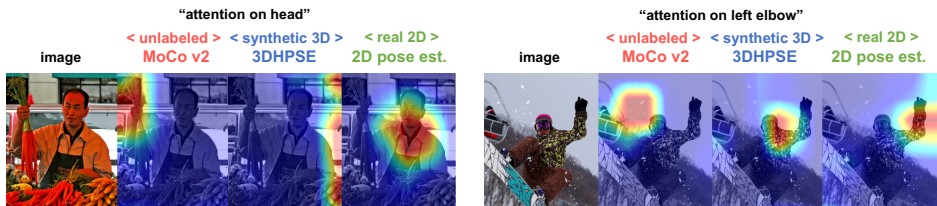

Figure 6: Feature attention visualization of PARE (Kocabas et al., 2021) for different human parts.

training in the aspect of labeling costs. In this regard, we answer two questions that could help future 3DHPSE researchers to choose their pre-training strategies.

(i) ***Does SSL entail a zero cost?*** No. It is not true in the aspect of data collection. Collecting, cleaning, and curating data for SSL demand resources (Russakovsky et al., 2015; He et al., 2019; Kotar et al., 2021). More importantly, the current SSL methods for 3DHPSE (Spurr et al., 2021; Zimmermann et al., 2021) assume a human to be centered in an input image, which is an unrealistic setting for in-the-wild images without bounding box labels. Thus, we should consider the labeling cost of bounding boxes when using SSL (Purushwalkam & Gupta, 2020; Goyal et al., 2021; El-Nouby et al., 2021). Further discussion is provided in Section B of Appendix.

(ii) ***Is SSL efficient than 2D annotation-based pre-training?*** No. SSL pre-trains on massive unlabeled data, which involves high computational costs. It often results in order of magnitude more computation than the supervised counterparts (Hénaff et al., 2021). In addition, due to double feature encoders and complex feature contrasting mechanism, SSL takes more time than 2D annotation-based pre-training to pre-train on the same amount of pre-training data. Pre-training ResNet-50 (He et al., 2016) with PeCLR (Spurr et al., 2021), MoCo v2 (Chen et al., 2020c), and the 2D pose estimator on MSCOCO took 110, 226, and 14 hours respectively, with the four RTX 2080Ti GPUs.

## 6 CONCLUSION

We have investigated different approaches for pre-training a 3DHPSE backbone. SSL, which has recently become the major trend in the community, has been thoroughly inspected. 2D annotation-based pre-training and synthetic data pre-training have also been experimented, since they share the similar motivation of reducing labeling costs and transferring useful representations to 3DHPSE. We experimented with multiple methods of each approach on multiple benchmarks to not draw a conclusion valid in a limited setting. Please also refer to Section D to confirm that our observations on the three pre-training approaches are preserved regardless of a 3DHPSE mesh regressor. Our empirical results show that 1) SSL is yet to replace the de facto paradigm of the ImageNet classification pre-training, and 2) 2D annotation-based pre-training can effectively transfer the knowledge to 3DHPSE, despite the least amount of pre-training data. We believe our findings, analysis, and discussion will significantly influence future research on pre-training for 3DHPSE.

**Acknowledgements.** This work was supported in part by the IITP grants [No.2021-0-01343, Artificial Intelligence Graduate School Program (Seoul National University), No.2022-0-00156, No. 2021-0-02068, and No.2022-0-00156], and the NRF grant [No. 2021M3A9E4080782] funded by the Korea government (MSIT), and AIRS Company in Hyundai Motor Company & Kia Corporation through HMC/KIA-SNU AI Consortium Fund.

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

APPENDIX

In this supplementary material, we provide more experiments and discussions that could not be included in the main text due to the lack of pages. The contents are summarized below:

A - Effects of SSL and 2D annotation-based pre-training on different amounts of data; InstaVariety(246K) (Kanazawa et al., 2019) and MPII(25K) (Andriluka et al., 2014) are experimented in addition to MSCOCO(150K) (Lin et al., 2014).

B - Discussion about labeling costs of SSL and 2D annotation-based pre-training

C - Effects of different pre-training approaches, when fine-tuned on different sizes of human data (30% and 60% of (Human3.6M (Ionescu et al., 2014) + MSCOCO (Lin et al., 2014)))

D - Effects of different human mesh regressors.

## A    EFFECTS OF PRE-TRAINING ON DIFFERENT AMOUNTS OF DATA

SSL is known to require large-scale pre-training data to be effective (He et al., 2020). In this regard, we apply PeCLR (Spurr et al., 2021) and MoCo v2 (Chen et al., 2020c) on InstaVariety(246K) (Kanazawa et al., 2019), which is $1.64\times$ bigger than MSCOCO(150K) (Lin et al., 2014). Interestingly, pre-training on InstaVariety consistently shows worse accuracy in all benchmarks as shown in Table 4. If we use less fine-tuning data, the accuracy gap is increased, except PeCLR on Human3.6M. We think the seemingly contradictory results come from the noisy bounding box used to crop a human from an image. The bounding box of a human in InstaVariety is calculated by measuring minimum and maximum $x, y$ locations of OpenPose (Cao et al., 2017)'s 2D pose estimation. Since the estimated values inevitably have errors, the bounding box obtained from them may not produce a human-centered image. Thus, the input distribution could be severely different during pre-training and fine-tuning, which could lead to counter-intuitive results.

We also provide the results of 2D pose estimation pre-training on MPII(25K) (Andriluka et al., 2014). Despite much less pre-training data, 2D pose estimation pre-training achieves comparable accuracy to that of pre-training MSCOCO. This shows the cost-effectiveness of 2D pose estimation pre-training, which is further discussed in the next section.

Table 4: Effects of pre-training on different amounts of data. InstaVariety(246K) (Kanazawa et al., 2019), MSCOCO(150K) (Lin et al., 2014), and MPII(25K) (Andriluka et al., 2014) are used for pre-training. The red and blue colors indicate the first and second best scores, respectively.

| fine-tuning data | pre-training data | pre-training method | 3DPW PA-MPJPE↓ | H36M PA-MPJPE↓ | MuPoTS 3DPCK↑ |
|---|---|---|---|---|---|
| H36M+ MSCOCO (100%) | InstaVariety (unlabeled) | PeCLR (Spurr et al., 2021) | 57.45 | 51.17 | 66.77 |
| | | MoCo v2 (Chen et al., 2020c) | 57.28 | 51.14 | 67.53 |
| | MSCOCO (unlabeled) | PeCLR (Spurr et al., 2021) | 57.27 | 49.84 | 66.88 |
| | | MoCo v2 (Chen et al., 2020c) | 56.35 | 50.06 | 66.95 |
| | MSCOCO | 2D pose est. | 53.34 | 44.89 | 69.04 |
| | MPII | 2D pose est. | 54.71 | 46.12 | 67.61 |
| H36M+ MSCOCO (10%) | InstaVariety (unlabeled) | PeCLR (Spurr et al., 2021) | 74.65 | 64.77 | 56.51 |
| | | MoCo v2 (Chen et al., 2020c) | 76.49 | 70.42 | 58.72 |
| | MSCOCO (unlabeled) | PeCLR (Spurr et al., 2021) | 71.24 | 66.58 | 60.16 |
| | | MoCo v2 (Chen et al., 2020c) | 66.93 | 62.06 | 61.36 |
| | MSCOCO | 2D pose est. | 56.92 | 55.31 | 67.12 |
| | MPII | 2D pose est. | 62.71 | 58.39 | 62.57 |

## B  DISCUSSION ABOUT LABELING COSTS OF SSL AND 2D ANNOTATION-BASED PRE-TRAINING

Some could value SSL on bounding box-labeled human data more than 2D annotation-based pre-training, despite the experimental results of the main text. For example, assuming less fine-tuning data like the main text's Table 6, MoCo v2 (Chen et al., 2020c) may be a more attractive option than the 2D pose estimation pre-training, considering that bounding box annotations take less cost than 2D pose annotations. In this context, we analyze the cost-effectiveness of different pre-training approaches by comparing the annotation cost in terms of annotation time.

Table 5 supports that the 2D pose estimation pre-training is much more cost-effective, given the similar annotation time. The state-of-the-art object annotation paper (Papadopoulos et al., 2017) reported 7 seconds per one bounding box annotation. We assume that the same time would take for a human bounding box. Liu et al. (Liu & Ferrari, 2017) reported 1.5 seconds per one human key point. Since a person in MPII (Andriluka et al., 2014) images have 14 key points, we can assume that the annotation time is 21 seconds per person. Cormier et al. (Cormier et al., 2021) reported 42.8 seconds per bounding box and pose of one person. We take the number of Cormier et al. (Cormier et al., 2021) in Table 5.

Table 5: Comparison between cost-effectiveness of SSL on bounding box-labeled data and 2D pose estimation pre-training.

| pre-training method | pre-training data | annotation time | 3DPW PA-MPJPE↓ | H36M PA-MPJPE↓ | MuPoTS 3DPCK↑ |
|---|---|---|---|---|---|
| MoCo v2 (Chen et al., 2020c) | MSCOCO (150K) (unlabeled) | 150K×7s=1050Ks | 56.35 | 50.06 | 66.95 |
| 2D pose est. | MPII (25K) | 25K×42.8s=1070Ks | 53.34 | 44.89 | 69.04 |

## C  FINE-TUNING ON DIFFERENT SIZES OF HUMAN DATA

We explore the effects of different pre-training approaches in different semi-supervised settings by varying sizes of fine-tuning data. Experimental results of using 10%, 30%, 60%, and 100% fine-tuning data of (Human3.6M (Ionescu et al., 2014) + MSCOCO (Lin et al., 2014)) are shown in Table 6 and Figure 7. SSL starts to surpass the random initialization baseline when fine-tuning data is reduced to 30%. Synthetic data pre-training also shows a similar tendency. Only 2D annotation-based pre-training consistently outperforms the random initialization baseline, and the accuracy margin is enlarged as the fine-tuning data reduces.

Table 6: 3DHPSE evaluation results on pre-training schemes with **different sizes of human data**. We report the best accuracy of each scheme on 3DPW, Human3.6M, and MuPoTS-3D dataset. The red and blue colors indicate the first and second best scores, respectively.

| fine-tuning data | pre-training data | pre-training method | 3DPW PA-MPJPE↓ | H36M PA-MPJPE↓ | MuPoTS 3DPCK↑ |
|---|---|---|---|---|---|
| H36M+ MSCOCO (100%) | - | random init. | 56.37 | 52.72 | 67.12 |
| | ImageNet (labeled) | classification | 55.65 | 48.36 | 67.76 |
| | MSCOCO (unlabeled) | PeCLR (Spurr et al., 2021) | 57.27 | 49.84 | 66.88 |
| | | MoCo v2 (Chen et al., 2020c) | 56.35 | 50.06 | 66.95 |
| | AGORA | 3DHPSE | 56.77 | 51.96 | 67.80 |
| | SURREAL | | 56.45 | 52.51 | 67.65 |
| | MSCOCO | DensePose est. | 54.16 | 49.18 | 68.43 |
| | | 2D pose est. | 53.34 | 44.89 | 69.04 |
| H36M+ MSCOCO (60%) | - | random init. | 57.31 | 50.61 | 67.30 |
| | ImageNet (labeled) | classification | 55.63 | 51.87 | 67.34 |
| | MSCOCO (unlabeled) | PeCLR (Spurr et al., 2021) | 59.17 | 52.14 | 65.80 |
| | | MoCo v2 (Chen et al., 2020c) | 58.57 | 52.34 | 66.63 |
| | AGORA | 3DHPSE | 57.23 | 53.99 | 68.87 |
| | SURREAL | | 57.69 | 53.10 | 66.47 |
| | MSCOCO | DensePose est. | 54.80 | 49.38 | 68.44 |
| | | 2D pose est. | 54.41 | 52.03 | 67.91 |
| H36M+ MSCOCO (30%) | - | random init. | 62.40 | 56.49 | 63.20 |
| | ImageNet (labeled) | classification | 57.85 | 53.02 | 65.74 |
| | MSCOCO (unlabeled) | PeCLR (Spurr et al., 2021) | 63.43 | 55.90 | 63.66 |
| | | MoCo v2 (Chen et al., 2020c) | 60.95 | 55.09 | 64.96 |
| | AGORA | 3DHPSE | 60.50 | 56.05 | 67.43 |
| | SURREAL | | 57.92 | 53.93 | 68.74 |
| | MSCOCO | DensePose est. | 55.44 | 52.37 | 68.40 |
| | | 2D pose est. | 53.94 | 47.39 | 68.56 |
| H36M+ MSCOCO (10%) | - | random init. | 73.37 | 67.59 | 57.32 |
| | ImageNet (labeled) | classification | 63.29 | 58.79 | 62.75 |
| | MSCOCO (unlabeled) | PeCLR (Spurr et al., 2021) | 71.24 | 66.58 | 60.16 |
| | | MoCo v2 (Chen et al., 2020c) | 66.93 | 62.06 | 61.36 |
| | AGORA | 3DHPSE | 64.71 | 63.17 | 65.73 |
| | SURREAL | | 62.51 | 59.10 | 64.50 |
| | MSCOCO | DensePose est. | 57.33 | 54.46 | 68.02 |
| | | 2D pose est. | 56.92 | 55.31 | 67.12 |

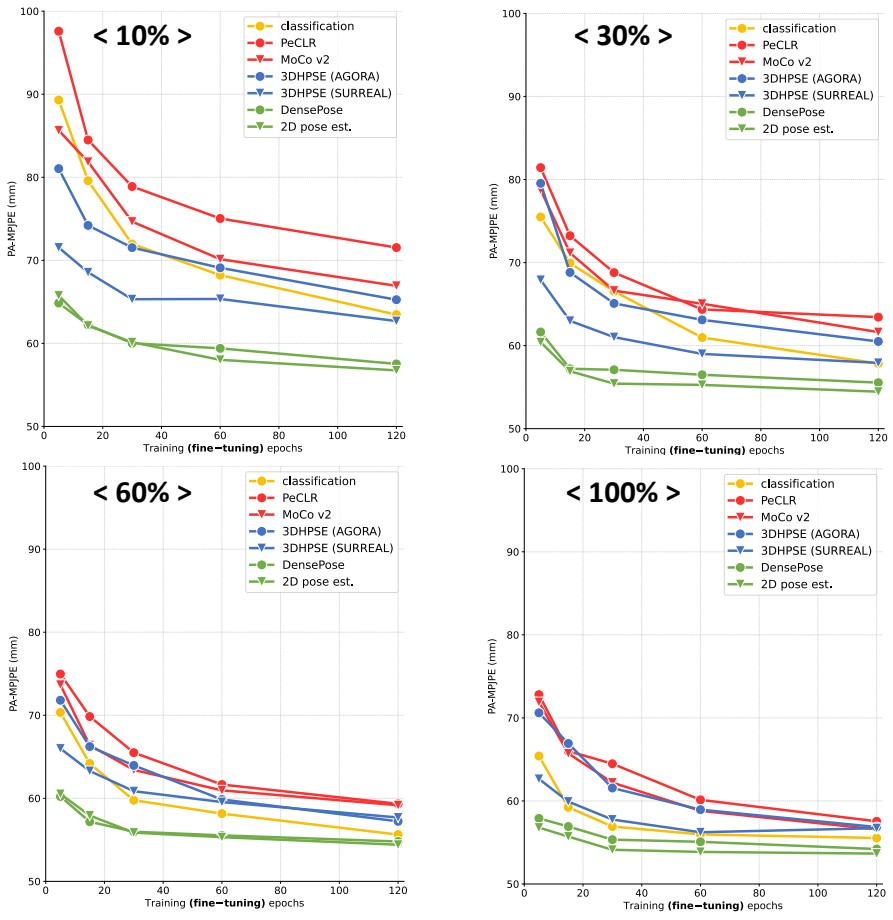

Figure 7: Learning curves of PA-MPJPE on 3DPW in the fine-tuning stage with different weights pre-trained on **different sizes of human data** by different pre-training methods.

# D EFFECTS OF DIFFERENT HUMAN MESH REGRESSORS

In Table 7 and Table 8, we experiment with different human mesh regressors (Kolotouros et al., 2019a; Zhang et al., 2021a; Moon et al., 2022a; Moon & Lee, 2020) that have distinct architectures and achieved state-of-the-art accuracy recently. The best-performing methods of SSL, synthetic data pre-training, and 2D annotation-based pre-training are used to compare. Overall, the same tendency is observed. SSL and synthetic data pre-training underperform the classification baseline, while 2D annotation-based pre-training outperforms it in multiple 3DHPSE benchmarks. This shows that our observations are not limited to a single 3DHPSE method but can generalize to different methods. The notable result is that I2L-MeshNet (Moon & Lee, 2020) produces overall the same accuracy regardless of the pre-training approaches. We conjecture that the different tendency of I2L-MeshNet comes from its architecture and target representation. While other human mesh regressors estimate the SMPL (Loper et al., 2015) parameters with MLP layers at the end of their networks, I2L-MeshNet directly regresses 2.5D locations of mesh vertices in a fully convolutional manner. The 2.5D representation expresses the $xy$ locations of mesh vertices in the image pixel space and the $z$ location as root joint-relative depth. As a result, the accuracy of I2L-MeshNet depends on the segmentation of a human from a cropped human-centric image. This partially posits the problem of 3DHPSE to the foreground estimation, which may less demand on learning complex priors about humans, such as 3D human body articulation.

Table 7: Effects of different human mesh regressors. We utilize SPIN (Kolotouros et al., 2019a), PyMAF (Zhang et al., 2021b), Pose2Pose (Moon et al., 2022a), and I2L-MeshNet (Moon & Lee, 2020) as regressors. We use full data (Human3.6M and MSCOCO) data for fine-tuning. The red and blue colors indicate the first and second best scores, respectively.

| regressor | pre-training data | pre-training method | 3DPW PA-MPJPE↓ | H36M PA-MPJPE↓ | MuPoTS 3DPCK↑ |
|---|---|---|---|---|---|
| SPIN | ImageNet | classification | 64.73 | 51.21 | 66.32 |
| | MSCOCO | MoCo v2 (Chen et al., 2020c) | 64.98 | 57.04 | 63.98 |
| | AGORA | 3DHPSE | 66.00 | 55.51 | 64.99 |
| | MSCOCO | 2D pose est. | 59.78 | 51.29 | 66.75 |
| PyMAF | ImageNet | classification | 60.37 | 67.23 | 68.19 |
| | MSCOCO | MoCo v2 (Chen et al., 2020c) | 64.09 | 69.49 | 65.49 |
| | AGORA | 3DHPSE | 65.60 | 71.63 | 66.72 |
| | MSCOCO | 2D pose est. | 59.04 | 69.53 | 70.46 |
| Pose2Pose | ImageNet | classification | 57.07 | 44.32 | 67.80 |
| | MSCOCO | MoCo v2 (Chen et al., 2020c) | 62.11 | 53.91 | 65.33 |
| | AGORA | 3DHPSE | 61.00 | 51.28 | 66.51 |
| | MSCOCO | 2D pose est. | 56.78 | 41.79 | 68.54 |
| I2L-MeshNet (lixel stage) | ImageNet | classification | 60.74 | 39.42 | 70.76 |
| | MSCOCO | MoCo v2 (Chen et al., 2020c) | 61.11 | 38.34 | 71.50 |
| | AGORA | 3DHPSE | 60.48 | 39.62 | 71.73 |
| | MSCOCO | 2D pose est. | 60.49 | 38.99 | 71.88 |

Table 8: Effects of different human mesh regressors in **semi-supervised setting**. We utilize SPIN (Kolotouros et al., 2019a), PyMAF (Zhang et al., 2021b), Pose2Pose (Moon et al., 2022a), and I2L-MeshNet (Moon & Lee, 2020) as regressors. We use 10% of (Human3.6M and MSCOCO) data for fine-tuning. The red and blue colors indicate the first and second best scores, respectively.

| regressor | pre-training data | pre-training method | 3DPW PA-MPJPE↓ | H36M PA-MPJPE↓ | MuPoTS 3DPCK↑ |
|---|---|---|---|---|---|
| SPIN | ImageNet | classification | 78.99 | 75.97 | 55.00 |
| | MSCOCO | MoCo v2 (Chen et al., 2020c) | 86.85 | 81.63 | 48.47 |
| | AGORA | 3DHPSE | 83.73 | 82.76 | 54.04 |
| | MSCOCO | 2D pose est. | 64.57 | 64.96 | 65.51 |
| PyMAF | ImageNet | classification | 67.64 | 73.62 | 63.76 |
| | MSCOCO | MoCo v2 (Chen et al., 2020c) | 71.90 | 80.43 | 58.69 |
| | AGORA | 3DHPSE | 69.65 | 81.71 | 62.04 |
| | MSCOCO | 2D pose est. | 59.41 | 73.84 | 68.91 |
| Pose2Pose | ImageNet | classification | 66.85 | 61.06 | 64.79 |
| | MSCOCO | MoCo v2 (Chen et al., 2020c) | 74.25 | 73.33 | 59.61 |
| | AGORA | 3DHPSE | 69.66 | 67.04 | 63.02 |
| | MSCOCO | 2D pose est. | 57.97 | 54.91 | 68.45 |
| I2L-MeshNet (lixel stage) | ImageNet | classification | 61.47 | 50.74 | 71.97 |
| | MSCOCO | MoCo v2 (Chen et al., 2020c) | 62.38 | 51.88 | 72.28 |
| | AGORA | 3DHPSE | 61.43 | 51.72 | 72.15 |
| | MSCOCO | 2D pose est. | 62.05 | 51.21 | 71.50 |

