# OpenReview forum: "Rethinking Self-Supervised Visual Representation Learning in Pre-training for 3D Human Pose and Shape Estimation"
_ICLR.cc/2023/Conference — ICLR 2023 poster_

### Official Review · Reviewer_EdAX · 2022-10-24

**Confidence:** 4
**Correctness:** 4
**Technical Novelty And Significance:** 3
**Empirical Novelty And Significance:** 3
**Recommendation:** 8

**Clarity, Quality, Novelty And Reproducibility:**

Overall, the clarity and quality of this paper are good. The experiments are also reproducible. As the conclusions of this paper bring limited effects on the community, the novelty is the main issue of this work.

**Strength And Weaknesses:**

Strength:

+ This paper is overall well-written and well-organized. The experimental results on different pre-training settings are convicing and support the claim of the paper.
+ A comprehensive study and analysis on the effectiveness of the pre-training for the 3DHPSE task is appreciated. The experiments on different self-supervised pre-training settings are new to the community, which is the aspect to support the acceptance of this paper.

Weaknesses:
- The main concerns regarding this submission are about its practical effects on the 3DHPSE community as the conclusion is actually in line with the practices of recent state-of-the-art methods. More specifically, recent approaches such as Pose2Pose and PARE have used the 2D pose pre-trained backbone in their training strategies. Different pretraining settings of the backbone are also discussed in [Zhang et al.].
Zhang et al., Learning 3D Human Shape and Pose From Dense Body Parts, TPAMI 2020

- The authors only investigate contrastive learning-based methods such as PeCLR for pose estimation. How about self-supervised learning methods on video data, for example [Sun et al. ECCV20, CVPR22]?
Sun et al., Self-Supervised Keypoint Discovery in Behavioral Videos, CVPR 2022
Sun et al., View-Invariant Probabilistic Embedding for Human Pose, ECCV 2020






**Summary Of The Paper:**

This paper conducted an empirical study on the effectiveness of the pre-training of a backbone for the task of 3D human pose and shape estimation (3DHPSE). In the experiments, the authors compared different pretraining strategies, including purely unlabeled self-supervised pre-training, 2D annotation-based pre-training, and synthetic data pre-training. The paper concludes that the self-supervised pre-training on unlabeled images is not effective for 3DHPSE and the 2D annotation-based pre-training brings the largest improvements.


**Summary Of The Review:**

This paper provides a comprehensive study on the effectiveness of the pre-training for the task of 3DHPSE, which is important to facilitate the research of the 3DHPSE task. The experiments are comprehensive and support the claims of the paper. However, the originality of this work is somewhat limited as parts of the claims have actually been adopted by recent state-of-the-art methods. Given the overall quality and originality, I currently rate this paper as marginally above acceptance.

---

> ### Author Response · Authors · 2022-11-13
> **Response to Reviewer EdAX**
>
> **EdAX-1. Practical effects on the 3DHPSE community.**
>
> We believe that this paper's practical effects on the 3DHPSE community are significant, as it first explores the unveiled effects of different pre-training approaches on 3DHPSE and provides solid pre-training baselines for 3DHPSE.
> It is common for researchers to adopt other research fields' successful techniques, such as SSL, and to apply them to their research.
> In this sense, there are growing interest and demand in applying SSL and other pre-training techniques to 3DHPSE.
> However, few papers have thoroughly analyzed the effects of SSL and other pre-training methods on 3DHPSE.
> For example, PARE only reported a faster convergence effect of 2D pose-based pre-training in the appendix, while we generally found accuracy increase of different 3DHPSE methods in different benchmarks after the 2D pose-based pre-training.
>
> In this regard, our empirical study practically benefits the 3DHPSE community by investigating the effects of pre-training with extensive experiments and bringing up valuable discussion points, such as the cost-effectiveness of SSL and 2D annotation-based pre-training.
> Also, we have thoroughly analyzed the reason behind the disappointing effects of SSL on 3DHPSE in Sections 4.2, 4.3, and Rebuttal s7cS-1.
> Future research on pre-training for 3DHPSE, especially regarding SSL, will refer to and benefit from our experiments, analysis, and discussions.
>
> **EdAX-2. Novelty over [Zhang et. al.].**
>
> Experiments and discussions on the different pre-training settings in [Zhang et. al.] [A] are fundamentally different from those of this paper.
> DaNet [A] decomposes the 3DHPSE pipeline to IUV map estimation from images and 3DHPSE from IUV maps.
> Figure 12 and Section 5.4 of DaNet are mainly discussing the effects of the IUV estimation quality on 3DHPSE.
> We think increasing the quality of the intermediate geometry representation (IUV map) is a separate research direction from transfer learning that pre-trains an image backbone.
> Specifically, the pre-training approaches covered in our paper are about representation learning to extract better _image features_ for 3DHPSE.
> Learning better image features is critical for 3DHPSE in that the approaches of leveraging intermediate geometry representation, such as Pose2Mesh and DaNet, lose valuable information in the image.
> For example, 3D human pose and shape outputs could be misaligned with images and not robust to occlusions, as discussed in 3DCrowdNet [B].
>
> **EdAX-3. SSL methods other than contrastive learning.**
>
> We experimented SwAV [C] in Tables 1 and 2, which is not based on contrastive learning among modern SSL methods.
> SwAV is an online clustering-based method that does not use a contrastive loss, but uses a "swapped prediction" mechanism to encode features from unlabeled images.
> The suggested video-based and multi-view setup-based SSL methods have limitations in generalization, although the research directions are interesting.
> [Sun et al., CVPR 2022] assumes a camera with a fixed location, while in-the-wild videos, such as videos of 3DPW, often have moving cameras.
> [Sun et al., ECCV 2020] requires data from a multi-view calibrated studio, which could limit the diversity of data (ex. poses and subjects).
> Also, the multi-view calibrated setting is infeasible in in-the-wild environments.
> Last, it uses 2D pose labels and thus cannot be categorized as SSL.
>
>
> [A] Zhang et. al. Learning 3D Human Shape and Pose From Dense Body Parts. TPAMI, 2020.
> [B] Hongsuk Choi, Gyeongsik Moon, JoonKyu Park, Kyoung Mu Lee, Learning to Estimate Robust 3D Human Mesh from In-the-Wild Crowded Scenes. In CVPR, 2022.
> [C] Caron et. al. Unsupervised Learning of Visual Features by Contrasting Cluster Assignments. In NeurIPS, 2020.

---

> ### Author Response · Authors · 2022-11-17
> **Follow-up**
>
> Dear reviewer,
>
> We would like to follow up to check if your concerns have been addressed. In the previous response, we made the following updates/clarification:
>
> - Regarding your concern about the practical effects on the 3DHPSE community, we re-emphasize the benefits of our work that provides solid pre-training baselines for 3DHPSE with in-depth analysis in **EdAX-1**. The differences with the mentioned prior works are also explained in **EdAX-1** and **EdAX-2**.
> - Regarding your suggestion about exploring other types of SSL other than contrastive learning, we clarified that SwAV is not a contrastive learning-based method and discussed why the suggested SSL methods could be inappropriate for our setting in **EdAX-3**.
>
> We are happy to answer further questions.

---

> > ### Comment · Reviewer_EdAX · 2022-12-03
> > **Thanks for the responses**
> >
> > The authors provided detailed responses, which did a good job to address most of my previous concerns. Though the 2D pose-based pre-training has been used in previous 3DHPSE methods, the comprehensive investigation of SSL on the task of 3DHPSE should be appreciated. After reading the comments of other reviewers and the author responses, I would like to recommend the acceptance of this paper.

---

> > > ### Author Response · Authors · 2022-12-05
> > > **Thank you for the response**
> > >
> > > Dear reviewer,
> > >
> > > We thank your response and appreciation of our work. We would like to ask whether the last comment indicates that the final recommendation score is going to be increased.
> > >
> > > Again, thank you for reading and responding to our responses.
> > >
> > > Paper2244 Authors

---

> > > > ### Comment · Reviewer_EdAX · 2022-12-06
> > > > **Follow-up response**
> > > >
> > > > Given the overall quality of this submission and the additional in-depth discussions provided by the authors, I would like to increase the recommendation score. I also agree with other reviewers that the analyses, discussions, and experiments mentioned in the author's responses should be included in the revised version of this paper.

---

> > > > > ### Author Response · Authors · 2022-12-07
> > > > > **Dear reviewer**
> > > > >
> > > > > We again thank your sincere review and suggestion. The additional experiments and discussions in our responses will be included in the revised version of this paper. The JointCon-related discussion in (Rebuttal s7cS-1) will be included in Section 4. The difference with the previous literature will be more elaborated in Section 2. Other analyses, discussions, and experiments mentioned in our responses will be also appropriately added to the manuscript.
> > > > >
> > > > > Thank you
> > > > > Paper2244 Authors

---

### Official Review · Reviewer_C9uS · 2022-10-24

**Confidence:** 5
**Correctness:** 3
**Technical Novelty And Significance:** 3
**Empirical Novelty And Significance:** 2
**Recommendation:** 6

**Clarity, Quality, Novelty And Reproducibility:**

The paper is clearly presented, and the investigation in SSL for human pose and shape estimation is new.

**Strength And Weaknesses:**


1. An in-depth investigation is presented in the paper to cover various pre-training methods.

2. A wide variety of base models have also been tested.


Weaknesses:
1. Although I appreciate this work as an investigation into using SSL for human pose and shape estimation, it might be difficult to justify the significance. Recent works have provided some studies on the impact of backbone selection for human pose and shape estimation [A, B], and training with 2D pose estimation as a proxy task has been shown to be effective for human pose and shape estimation.

2. It is unconventional to use backbone pretraining + head fine-tuning for human pose and shape estimation as fully supervised training has been widely adopted in the field. I agree that it is interesting if pretraining has shown promising results, but there is a significant gap between the evaluated pretraining methods and the SoTA performance. E.g. PARE achieves easily <50 mm PA-MPJPE on 3DPW, but struggles at ~55 mm PA-MPJPE in Table 1. The practical use of this investigation might be questionable.

3. The conclusion on synthetic data might need some adjustments. It has been shown in [C, B] that synthetic data, if used properly, improve baseline models such as HMR, SPIN, VIBE and PARE on real datasets.

4. The semi-supervised learning setting might not be very relevant in today's context as synthetic data incurs almost no cost to scale up. In addition, semi-supervised learning results in a performance gap that is too significant to easily justify its usefulness in real-life applications.

5. There seems to be a slight contradiction in Sec 4.2. It is stated that "a backbone network is preferred to learn more about human features" so SSL does not work, but it is also stated that "high-level semantics ... could be beneficial transferred to inference on humans". I think it may be helpful to provide a more clear-cut explanation that separates classification pretraining and SSL.

[A] Pang et. al., Benchmarking and Analyzing 3D Human Pose and Shape Estimation Beyond Algorithms, NeurIPS (Dataset Track) 2022

[B] Cai et. al., Playing for 3D Human Recovery, arXiv 2021

[C] Patel et. al. AGORA: Avatars in Geography Optimized for Regression Analysis, CVPR 2021


**Summary Of The Paper:**

This work provides a thorough evaluation of various backbone pre-training methods such as SSL, with human datasets, and semi-supervised setting.

**Summary Of The Review:**

Despite I deeply appreciate the investigation in pretraining backbone with SSL for human pose and shape estimation, I find it difficult to justify the significance as the finding seems to suggest that SSL does not work in this case. More importantly, there are relevant works on backbone training strategies and synthetic data, showing much more promising results. Therefore, I find the paper in its current form does not meet the bar of ICLR.

---

> ### Author Response · Authors · 2022-11-13
> **Response to Reviewer C9uS (Part 2)**
>
> **C9uS-4. Synthetic dataset.** - continued
>
> One fine-tuning strategy of [B] is a mixed-batch training of synthetic and real data.
> The mixed-batch training is to train a network with a mini-batch composed of image samples from different datasets that have different domains.
> We applied the mixed-batch training in the pre-training by pre-training a backbone with AGORA and MSCOCO.
> 3D human pose and shape supervision is given to the AGORA image samples and 2D pose supervision is given to the MSCOCO image samples.
> The mixed pre-training achieves 52.36mm/56.92mm on 3DPW, 49.54mm/56.40mm on Human36M, and 69.04/67.09 3DPCK on MuPoTS, when fine-tuned on 100\%/10\% data (Human36M + MSCOCO).
> Compared to solely pre-training on AGORA, it greatly improves accuracy in all three benchmarks.
> Compared to the 2D pose-based pre-training on MSCOCO, it improves the accuracy on 3DPW by 0.98mm/0.94mm PA-MPJPE on 3DPW, degrades the accuracy on Human36M by 4.65mm/1.09mm PA-MPJPE, when fine-tuned on 100\%/10\% data (Human36M + MSCOCO).
> The comparisons are based on the results of Table 2.
> The results show that the effectiveness of synthetic data pre-training is valid when used with real data (MSCOCO), and do not change our observation that 2D annotation-based pre-training can effectively transfer the knowledge to 3DHPSE.
> In addition, it does not change our conclusion that SSL is yet to replace the de facto paradigm of the ImageNet classification pre-training.
>
>
> **C9uS-5. Semi-supervised learning setting.**
>
> The semi-supervised learning setting is a more practical setting for other research and industrial fields, such as medical applications and robotics, that use 3DHPSE.
> In such cases, off-the-shelf pre-trained 3DHPSE methods cannot be directly used, due to unsatisfying performance in their domain-specific images.
> As a result, it is a common practice for developers and researchers to collect a small set of their domain-specific real data and fine-tune the 3DHPSE method on it.
> For example, in the behavioral study and clinical examination of children, data with child-specific appearances and poses is necessary, which is rare and highly differs from the existing data.
> This is one of the reasons why many representation learning papers (Chen et al., 2020a;b, Caron et al., 2020, Spurr et al., 2021) provide results of the semi-supervised learning setting.
>
> Also, generating high-quality synthetic data is often costly and challenging to scale up, although it is still more affordable and feasible than collecting real 3D data.
> For example, gathering high-resolution 3D scans of diverse human subjects requires various resources like professional actors and a multi-view calibrated studio.
> Imitating the diverse real-world human motions in the simulator is another unresolved research problem, since in-depth knowledge of physics and high-level simulating techniques are necessary to implement precise human-human interaction, object manipulation by humans, penetration prevention, and so on.
>
> Thus, the large performance gap in the semi-supervised learning setting strengthens our empirical study's significance in the real life.
>
> **C9uS-6. Difference between classification pre-training and SSL.**
>
> The effects of the classification pre-training and SSL on ImageNet are different, since the classification pre-training can learn high-level priors about objects while the current SSL is hard to.
> The high-level priors, such as the understanding about the structures of objects, are encouraged to learn by the classification supervision.
> Each of the priors is useful partial information for making a later classification decision (Yosinski et al., 2015) [G].
> And the high-level priors about objects can be generalized to different classes.
> For example, Figure 2 of [G] shows that the same layer of the ImageNet classification pre-trained backbone activates appropriate regions of human and lion faces, even though ImageNet does not have any face class.
>
> The current SSL differentiates the representations of instances in the same class (Khosla et al., 2020), and thus it is hard to learn the high-level priors that are common in one or more classes.
> Please refer to Rebuttal s7cS-1 for more explanation.
>
> [A] Pang et. al. Benchmarking and Analyzing 3D Human Pose and Shape Estimation Beyond Algorithms. In NeurIPS (Dataset Track), 2022.
> [B] Cai et. al. Playing for 3D Human Recovery. arXiv, 2021.
> [C] Patel et. al. AGORA: Avatars in Geography Optimized for Regression Analysis. In CVPR, 2021.
> [D] Kocabas et. al. PARE: Part attention regressor for 3D human body estimation. In ICCV, 2021.
> [E] Gyeongsik Moon, Hongsuk Choi, and Kyoung Mu Lee. Accurate 3D Hand Pose Estimation for Whole-Body 3D Human Mesh Estimation. In CVPR workshop, 2022.
> [F] Zhu et. al. Unpaired image-to image translation using cycle-consistent adversarial networks. In ICCV, 2017.
> [G] Yosinski et. al. Understanding neural networks through deep visualization. In ICML workshop, 2015.

---

> ### Author Response · Authors · 2022-11-13
> **Response to Reviewer C9uS (Part 1)**
>
> **C9uS-1. Significance of work.**
>
> We thank the reviewer's valuable comments.
> While [A] provides some experimental results of three pre-training approaches (classification on ImageNet, 2D pose est. on MPII, 2D pose est. on MSCOCO), the effects of SSL and synthetic data are still unexplored.
> In particular, considering that SSL has shown a powerful impact on other vision tasks, the extensive experiments and analysis on the current SSL methods distinguish our work from [A].
> More importantly, [A] is a concurrent work according to the ICLR 2023 reviewer guideline (https://iclr.cc/Conferences/2023/ReviewerGuide).
> [B] studies the selection of a backbone from different architectures (ResNet and DeiT), but it does not cover the pre-training approaches in a given architecture.
> In contrast to [A, B], we provide in-depth and extensive experimental results for pre-training approaches of 3DHPSE.
> As well as the conclusion of the best pre-training method, our paper includes the reasoning about the current SSL's disappointing performance and a discussion about the cost-effectiveness of pre-training methods.
> Our in-depth discussions will impact academic and industrial fields' future decisions on data collection for 3DHPSE and method development on 3DHPSE pre-training.
>
> On the one hand, recent methods use a proxy task as 2D pose estimation, and the proxy task is performed in the fine-tuning stage.
> Our work focuses on a pre-training strategy, which is also an important factor impacting the 3DHPSE performance.
> As shown in Table 3., Pose2Pose [E] using a proxy task with 2D pose estimation is highly affected by the pre-training strategies.
> Unlike the widely studied fine-tuning strategy, such as using a proxy task, pre-training is still an unexplored area, and our paper provides in-depth observations of the pre-training.
>
>
> **C9uS-2. Convention of 3DHPSE training.**
>
> In the fine-tuning stage, both of backbone and head are trained in an end-to-end manner (3rd paragraph of Section 3.).
> The pre-training-and-fine-tuning strategy is a conventional training procedure in most 3DHPSE methods (1st paragraph of Section 1. and 2nd paragraph of Section 3.).
> Considering the pre-training strategy is commonly adopted and has a significant impact on 3DHPSE performance, the extensive study on pre-training is vital and valuable for the 3DHPSE community.
>
>
> **C9uS-3. Validity of experiment results.**
>
> As the reviewer mentioned, PARE achieves SOTA performance, 46.5 mm (< 50 mm) PA-MPJPE on 3DPW, while there are two differences with our setting.
> First, PARE additionally use 3DPW during fine-tuning, but our setting does not use 3DPW.
> Since the domains of the train and test set of 3DPW are similar, an experimental protocol that does not use 3DPW for the training is mainly used in the 3DHPSE community.
> Second, PARE use HRNet-w32 as a backbone, but our setting use ResNet-50, which is more commonly used in most 3DHPSE methods.
> Considering the above two, [D] reported that the evaluation results of PARE without the 3DPW training set and with ResNet-50 is 52.3 mm PA-MPJPE on the 3DPW test set.
> Our reported accuracy when pre-training with 2D pose est. is 53.34 mm, which is a reasonable performance.
> The slight difference in accuracy is from the difference in training datasets: [D] uses 5 datasets (Human36M, MSCOCO, MPII, LSPET, MPI-INF-3DHP), and we use 2 datasets (Human36M, MSCOCO) for training PARE.
> In this regard, when using additional datasets in our setting (Human36M, MSCOCO, MPII, MPI-INF-3DHP), we can observe PARE achieve the accuracy of 52.05 mm PA-MPJPE on 3DPW, which is similar to the reported by PARE.
>
>
> **C9uS-4. Synthetic dataset.**
>
> We do not conclude that synthetic data pre-training is not beneficial for 3DHPSE.
> It achieves better accuracy than SSL and also outperforms the ImageNet classification pre-training in the semi-supervised setting.
> [B, C] experiments on the effects of fine-tuning with synthetic data, which is a separate research direction from our empirical study that validates the effectiveness of different data types in the representation learning aspect.
> The CycleGan [F]-based pre-training in [B] is about domain adaptation that transforms synthetic images into realistic images, and it is far from representation learning.
> Also, the CycleGan [F]-based pre-training is introduced in the latest version of [B], which is a concurrent work according to the ICLR 2023 reviewer guideline (https://iclr.cc/Conferences/2023/ReviewerGuide).

---

> ### Author Response · Authors · 2022-11-17
> **Follow-up**
>
> Dear reviewer,
>
> We would like to follow up to check if your concerns have been addressed. In the previous response, we made the following updates/clarification:
>
> - Regarding your concern about the significance of this work, we clarified our novel contribution against prior works including [A,B] in **C9uS-1**. Very few prior works have focused on representation learning for 3DHPSE. Our extensive experiments with a variety of pre-training settings and the analysis will significantly benefit the 3DHPSE community.
> - Regarding your concern about 'backbone pretraining + head fine-tuning', we clarified that both backbone and head are trained in an end-to-end manner in the fine-tuning stage in **C9uS-2**. Also, we assured the validity of our experiment results by explaining the accuracy gap regarding PARE in **C9uS-3**.
> - Regarding your concern about the conclusion on synthetic data, we clarified our conclusion and difference with [B,C] in **C9uS-4**. We performed a new experiment that adopts the mixed-batch training of [B], but the results did not change the conclusion.
> - Regarding your concern about the semi-supervised setting, we emphasized the importance of the semi-supervised setting and discussed the cost of synthetic data in **C9uS-5**.
> - We provided a more clear explanation of the different effects between classification pre-training and SSL for 3DHPSE in **C9uS-6**.
>
> We are happy to answer further questions.

---

> > ### Comment · Reviewer_C9uS · 2022-12-06
> > **Thanks for the response**
> >
> > I would like to thank the authors for the response that addresses most of my concerns. The manuscript will be significantly improved if the following points are added in the revised version:
> >
> > - A throughout explanation and the new approach JointCon on "why SSL doesn't work" as in s7cS-1. I believe this investigation serves to fill the current gap of the study on SSL for 3DHPSE.
> >
> > - Additional experiments including PARE with more conventional five training datasets that achieves improvement on top of the strong baseline (C9uS-3), and mixed training with synthetic data (C9uS-4).
> >
> > - Discussion (C9uS-1) of existing works that leverage 2D cues and synthetic data for 3DHPSE for completeness.
> >
> > Please let me know your thoughts.
> >
> > In addition, is it correct to conclude that current SSL strategies are insufficient to address 3DHPSE, and more explored ways of adding data (e.g., 2D supervision & synthetic data) is still the way to go?

---

> > > ### Author Response · Authors · 2022-12-06
> > > **Thank you for the response**
> > >
> > > We sincerely thank your constructive feedback.
> > >
> > > We agree that the suggested points will improve the value and completeness of this work.
> > > - We will enhance the discussion in Section 4.2 with the new approach JointCon (s7cS-1) and analysis of it.
> > > - We will add the experiments of PARE with the suggested training settings, including the experimental results in (C9uS-4).
> > > - We will include the discussion (C9uS-1) of existing works in the related work Section 2.
> > >
> > > Last, we believe it is hard to conclude which data (e.g., unlabeled data, 2D supervision data, and synthetic data) is more worth collecting for 3DHPSE. Instead, we think our work shows the importance and potential of different data type-based _pre-training_ for 3DHPSE, which has been under-explored in the literature. From the discussions in Section 4.2 and (s7cS-1), we think it could be better to make SSL learn high-level priors (e.g., object structure) for 3DHPSE in the future. Other data type-based pre-training also needs more exploration. We think JointCon-like pre-training methods could be more improved and developed in the future. The mixed pre-training with synthetic data could be also an interesting direction. Future research on pre-training for 3DHPSE, especially regarding SSL, will refer to and benefit from our experiments, analysis, and discussions.
> > >
> > > Again, thank you for responding to our responses and for the valuable suggestions.
> > >
> > > Paper2244 Authors

---

> > > > ### Comment · Reviewer_C9uS · 2022-12-06
> > > > **Updated recommendation**
> > > >
> > > > The current state of the manuscript has achieved significant improvements from its original state. Hence, I would like to update my score accordingly.
> > > >
> > > > I would like to thank the authors for the discussion, and it may be helpful to include it in the conclusion of the manuscript.

---

> > > > > ### Author Response · Authors · 2022-12-07
> > > > > **Dear reviewer**
> > > > >
> > > > > We again appreciate your sincere and constructive feedback. The discussion was indeed helpful to improve the manuscript. We will reinforce the conclusion and add the three points suggested.
> > > > >
> > > > > Paper2244 Authors

---

### Official Review · Reviewer_jtxR · 2022-10-24

**Confidence:** 5
**Clarity, Quality, Novelty And Reproducibility:** The presentation and experiment desig…
**Correctness:** 4
**Technical Novelty And Significance:** 2
**Empirical Novelty And Significance:** 3
**Recommendation:** 6

**Strength And Weaknesses:**

Strengths:
1. The paper is easy to follow and well-written. The authors described their experiments and different analysis settings clearly.
2. The authors conducted very thorough experiments to validate which data is useful for pre-training the backbone.
3. The paper provides several interesting discussions and insightful findings. It could be a useful reference for future research.

Weaknesses:
1. Prior works have been discovered that 2D re-projection loss is very critical to exiting methods. The main message of this paper seems to double confirm the importance of 2D pose data. While the authors show that pre-training the backbone is important, there is no other new pre-training technique introduced to effectively boost the SOTA performance.
2. I appreciate the efforts of experimenting with SSL methods. However, the results seem not very positive. Since SSL might be a bit more difficult to train, I wonder if the SSL pre-training needs more training tricks and more diverse data to enhance feature representation generalizability.

**Summary Of The Paper:**

This paper provides empirical study of how to pre-train the vision backbone for human pose and mesh estimation. The authors conducted many experiments with different pre-training data. The authors observed using real images with 2D human pose labels (such as COCO) is the most effective one.

**Summary Of The Review:**

I have a mixed feeling about this paper. The paper presents a very useful exploration of pre-training vision backbone for human pose and mesh reconstruction. All the discussions and analysis are valuable. It will be very helpful to future researchers. This is good as a technical report. But when considering it as a research paper, the novelty of pre-training technique is weak.

Probably because the considered SSL pre-training methods don't work well, the paper consists of several negative observations for SSL. The only positive signal comes from pre-training with real images & pose labels (such as COCO dataset). This is more like a revisit, as some recent works have pre-trained a PoseNet in a similar way for pose-to-mesh pipeline. That is, there is no novel pre-training technique introduced. This weakens the contribution of this submission.

However, considering the valuable analysis, discussions, and the great efforts of conducting all these experiments, this paper still provides useful guidance to the community.

---

> ### Author Response · Authors · 2022-11-13
> **Response to Reviewer jtxR (Part 2)**
>
> **jtxR-2. Negative observations on SSL.** - continued
>
> Two different versions of JointCon are experimented.
> The first version contrasts joint-level features of the same joint class from different images as other SSL methods (Chen et al., 2020b; Chen & He, 2020; Chen et al., 2020c; He ́naff et al., 202, Spurr et al., 2021).
> The 'positive' samples of an anchor joint-level feature are only from augmentations, and joint-level features of the different joint classes and the same joint class from different images are treated as 'negative' samples.
> The second version treats joint-level features of the same joint class from different images as 'positive samples', inspired by the supervised contrastive learning [B].
> The first version of JointCon achieves 56.71mm/59.59mm PA-MPJPE on 3DPW, 47.99mm/56.55mm PA-MPJPE on Human36M, and 67.85/67.38 3DPCK on MuPoTS, when pre-trained on MSCOCO and fine-tuned on 100\%/10\% data (Human36M + MSCOCO).
> The second version of JointCon achieves 54.25mm/57.19mm PA-MPJPE on 3DPW, 45.52mm/54.97mm PA-MPJPE on Human36M, and 68.87/67.98 3DPCK on MuPoTS, when pre-trained on MSCOCO and fine-tuned on 100\%/10\% data (Human36M + MSCOCO).
> The JointCon's second version outperforms the first version in all three benchmarks.
>
> The accuracy gap between the two versions of JointCon proves that it is important to learn consistent representations across instances of the same class.
> Contrasting representations of the instances in the same class (i.e. objects or joints in the same class) hinders a backbone from learning useful priors and degrades the performance of 3DHPSE.
> The ImageNet classification pre-training, 2D annotation-based pre-training, and synthetic data pre-training rather learn consistent representations of the instances in the same class and achieve better accuracy than SSL.
> The other important observation is that both JointCon versions outperform PeCLR, MoCo, and SwAV on all benchmarks in Table 2.
> The accuracy boost by only adding joint information to the SSL system demonstrates that the current SSL's poor performance is not coming from architecture, data-augmentation, or losses, but from the instance-level learning characteristic.
>
> We believe the results of the SSL methods in our paper are firm and solid.
> In the experiments of Table 1, the pre-trained backbone weights are directly imported from the papers' official code repositories, which maintain the best checkpoints of their models.
> In the experiments of Table 2, we adapted SSL methods to human datasets, but the core pre-training code including augmentation is unchanged from their official code.
> We referred to PeCLR and Hanco, which are SSL methods adapted to human hands, to set the batch size and the pre-training epochs.
>
> Last, we have experimental results of SSL methods on InstaVariety (246K), which is 1.64 times bigger than MSCOCO (150K), in Section A and Table 4.
>
> **jtxR-3. Revisit to pre-training a PoseNet.**
>
> A few recent 3DHPSE works used the 2D pose-based pre-training, but its true effects were not unveiled.
> For example, PARE only reported faster convergence in the appendix, while we generally found accuracy increase of different 3DHPSE methods in different benchmarks after 2D pose-based pre-training.
> The observations and analysis in our paper will make the comparison between different 3DHPSE methods more fair and clear.
> Also, discussions on the cost-effectiveness and pre-training efficiency of SSL and 2D annotation-based pre-training (Sections 5 and B) will impact academia and industrial fields' future decisions on data collection and method development.
>
> Pose2Mesh [D] pre-trained a PoseNet that estimates a 3D pose from a 2D pose.
> Its pre-training essentially differs from the pre-training approaches covered in this paper.
> It is more like leveraging the advantage of intermediate geometric representation (3D pose) that can break down and ease the complexity and ambiguity of 3DHPSE.
> On the other hand, our interest in pre-training is in learning useful visual representations that can be transferred to 3DHPSE.
>
> [B] Khosla et. al. Supervised contrastive learning. In NeurIPS, 2020.
> [D] Hongsuk Choi, Gyeongsik Moon, and Kyoung Mu Lee. Pose2Mesh: Graph convolutional network for 3D human pose and mesh recovery from a 2D human pose. In ECCV, 2020.

---

> ### Author Response · Authors · 2022-11-13
> **Response to Reviewer jtxR (Part 1)**
>
> **jtxR-1. Novelty.**
>
> We thank the reviewer's appreciation of the paper's analysis, discussions, and efforts in conducting the experiments.
> As an empirical study like [A], our paper's novelty lies in the new observations on pre-training for 3DHPSE.
> In the 3DHPSE literature, this paper is the first to visit the importance of different data types in the representation learning aspect, by thoroughly comparing and analyzing the effects of SSL methods and other diverse pre-training approaches.
> We think our discussions, including the reasoning about the current SSL's disappointing performance for 3DHPSE (Sections 4.2 and 4.3) and the SSL's cost-effectiveness (Sections 5 and B), will impact many researchers when deciding to collect data or develop a new SSL method.
>
> Also, we additionally experimented a new pre-training approach, _JointCon_, introduced in s7cS-1 and below to further investigate the reason behind the current SSL's poor performance for 3DHPSE.
>
> **jtxR-2. Negative observations on SSL.**
>
> We think SSL, based on unlabeled data, is still a valuable pre-training approach, when fine-tuning data is scarce.
> As shown in Tables 1, 2, and 6, MoCo v2 successfully outperforms the random initialization baseline in the semi-supervised setting.
>
> There are two main reasons for the disappointing results of the current SSL.
> First is that representations learned by the current SSL are inconsistent across instances of the same class (Khosla et al., 2020) [B], as discussed in Section 4.2: Question (ii).
> The representations of instances in the same class are pushed apart if the instances are from different images, which results in inconsistency.
> Due to the representations' inconsistency, the SSL's backbone has difficulty in learning high-level priors about a specific class, the human, such as features of human parts, regardless of the data types (arbitrary objects data and human data) it pre-trains on.
> On the contrary, the representations learned by the ImageNet classification pre-training embed high-level priors about a specific class, and the priors could be also interchangeably leveraged across different classes.
> The high-level priors include the understanding about the structures of objects, which are useful partial information for making a later classification decision (Yosinski et al., 2015) [C].
> Figure 2 of [C] shows that one layer of the ImageNet classification pre-trained backbone can activate appropriate regions of human faces, even though ImageNet does not have any face class nor a human class.
>
> The other main reason is the inherent task gap between the current SSL and 3DHPSE.
> A 3D human pose, one of the targets of 3DHPSE, is a complex set of multiple joints that relates to each other in the kinematic chain.
> However, the current methods of SSL have an instance-level learning characteristic, which is unaware of joints that add up to a human pose.
> They augment an instance (human)-centered image and try to encode a common feature from the augmented and unaugmented images.
> Their mechanism could make an image backbone learn visual representations that could generalize well to instance-level downstream tasks, such as classification and object detection.
> However, the visual representations essentially lack understanding of the fine-level semantic information, the human joints.
>
> To empirically validate the two reasons discussed above, we designed a new pre-training approach that combines an SSL approach with 2D joint labels (2D pose).
> We call it _JointCon_.
> It applies contrastive learning to joint-level features, which are locally sampled image features using GT 2D joint locations following Pose2Pose (Moon et al., 2022a).
> The new approach shares the same network architecture, contrastive loss, and augmentations with PeCLR (Spurr et al., 2021).
>
> [A] Kaiming He, Ross Girshick, and Piotr Dollar. Rethinking imagenet pre-training. In ICCV, 2019.
> [B] Khosla et. al. Supervised contrastive learning. In NeurIPS, 2020.
> [C] Yosinski et. al. Understanding neural networks through deep visualization. In ICML workshop, 2015.

---

> ### Author Response · Authors · 2022-11-17
> **Follow-up**
>
> Dear reviewer,
>
> We would like to follow up to check if your concerns have been addressed. In the previous response, we made the following updates/clarification:
>
> - Regarding your concern about the novelty, we clarified our empirical novelty and significance against the previous literature in **jtxR-1** and **jtxR-3**. Also, we provided a new pre-training technique, _JointCon_, to further investigate the reason behind the current SSL's negative performance for 3DHPSE in **jtxR-2**.
> - Regarding your concern about the negative outcomes of the current SSL, we provided a more in-depth investigation to explain the current SSL's limitations for 3DHPSE with experiments with JointCon in **jtxR-2**. The experimental results and analysis are in line with those in Sections 4.2 and 4.3 and will be included in the final version of this paper. Also, we assured that the SSL pre-training methods in our experiments followed their best practice in  **jtxR-2**.
>
> We are happy to answer further questions.

---

### Official Review · Reviewer_s7cS · 2022-10-24

**Confidence:** 4
**Correctness:** 3
**Technical Novelty And Significance:** 2
**Empirical Novelty And Significance:** 4
**Recommendation:** 8

**Clarity, Quality, Novelty And Reproducibility:**

In general the paper is well-written and easy to read with the main findings of the paper clearly presented and articulated.

One point of confusion for me is:

- The semi-supervised setting is referred to many times in the main text and forms one of the main talking points in section 4.2. I may have missed it, but it is not entirely clear how semi-supervised learning is implemented in this paper.

**Strength And Weaknesses:**

Strengths:

- The paper presents a very nice set of experiments with many settings which I think would be very much appreciated by those working on the problem of 3D human pose estimation.

Weaknesses:

- There is no real investigation as to why SSL does not work as well for pre-training. There is some conjecture in section 4.2 but not much probing experimentally or qualitatively beyond the pre-training conditions and the final performance result. But there is no probing or analyzing the representations learnt in the different set-ups. For example networks trained on ImageNet in a supervised fashion are great at recognising textures whereas self-supervised representations are not good at recognising color. Are such issues coming into play? Is the problem the data-augmentations being used as opposed to SSL? Which augmentations could be problematic and why?

**Summary Of The Paper:**

This paper presents an exhaustive empirical study into the effect on performance, for 3D human pose and shape estimation, of different types of pre-training of the backbone network using different training datasets. The different types of pre-training investigated are supervised classification, SSL, and semi-supervised learning and the datasets used vary from large and diverse (ImageNet) to human centric (MSCOCO)  and synthetic.

The main conclusions from the empirical results are for 3D human pose and shape estimation:
1) supervised ImageNet pre-training dominates by quite a large margin the standard SSL ImageNet pre-training approaches
2) Pre-training on smaller but more domain specific tasks (2d keypoint estimation) and data (images of people) can be as effective as pre-training on larger but more diverse datasets.

Contributions:

* The paper presents a very thorough and extensive  set of sensible experiments exploring if SSL pre-training should replace the standard practice of using a backbone pre-trained on ImageNet in a supervised fashion for 3D human pose and shape estimation and deciding it should definitely not as this point in time!

**Summary Of The Review:**

The empirical results in this paper would be of interest to many. But perhaps, because of the lack of insightful analysis and probing into the reason for the poor performance of SLL pre-training this would cause some hesitation in acceptance at ICLR.

After the authors' response:

I appreciate the authors' detailed response to the review and the explanation for the "relative" poor performance of SSL pre-training. After reading the response I have increased my rating to an accept.

---

> ### Author Response · Authors · 2022-11-13
> **Response to Reviewer s7cS (Part 2)**
>
> **s7cS-1. In-depth investigation on the poor performance of SSL pre-training.** - continued
>
> The accuracy gap between the two versions of JointCon proves that it is important to learn consistent representations across instances of the same class.
> Contrasting representations of the instances in the same class (i.e. objects or joints in the same class) hinders a backbone from learning useful priors and degrades the performance of 3DHPSE.
> The ImageNet classification pre-training, 2D annotation-based pre-training, and synthetic data pre-training rather learn consistent representations of the instances in the same class and achieve better accuracy than SSL.
> The other important observation is that both JointCon versions outperform PeCLR, MoCo, and SwAV on all benchmarks in Table 2.
> The accuracy boost by only adding joint information to the SSL system demonstrates that the current SSL's poor performance is not coming from architecture, data augmentation, or losses, but from the instance-level learning characteristic.
>
> Last, Figure 6 in the main paper visualizes each human part’s feature activation of PARE (Kocabas et al., 2021) to show how representations learned by each pre-training approach affect the human mesh regressor’s understanding of human geometry.
>
> **s7cS-2. Semi-supervised setting.**
>
> To evaluate the effects of different pre-training approaches when fine-tuning data is scarce, we used 10\% of Human36M and 10\% of MSCOCO data for the semi-supervised setting as written at the bottom of page 4 and page 7.

---

> ### Author Response · Authors · 2022-11-13
> **Response to Reviewer s7cS (Part 1)**
>
> **s7cS-1. In-depth investigation on the poor performance of SSL pre-training.**
>
> The data augmentation difference between SSL and 3DHPSE is not the main reason for the poor performance of SSL pre-training.
> For example, PeCLR (Spurr et al., 2021) adapted the previous SSL's geometric augmentations of unlabeled images (scale, rotation, translation) to 3D hand pose and shape estimation during SSL pre-training.
> The core intuition of the adaptation is to make features of hand images equivariant to the geometric augmentations, since the targets (i.e. 3D hand pose and shape) are equivariant to the geometric augmentations.
> The intuition is the same in 3DHPSE, and we applied PeCLR in 3DHPSE with their code.
> However, it is only effective when significantly less fine-tuning data (10\% Human3.6M + 10\% MSCOCO, < 50K) is used, as shown in Table 2 and discussed in Section 4.3.
>
> One of the main reasons for the poor performance of SSL pre-training is that representations learned by the current SSL are inconsistent across instances of the same class (Khosla et al., 2020) [A], as discussed in Section 4.2: Question (ii).
> The representations of instances in the same class are pushed apart if the instances are from different images, which results in inconsistency.
> Due to the representations' inconsistency, the SSL's backbone has difficulty in learning high-level priors about a specific class, the human, such as features of human parts, regardless of the data types (arbitrary objects data and human data) it pre-trains on.
> On the contrary, the representations learned by the ImageNet classification pre-training embed high-level priors about a specific class, and the priors could be also interchangeably leveraged across different classes.
> The high-level priors include the understanding about the structures of objects, which are useful partial information for making a later classification decision (Yosinski et al., 2015) [B].
> Figure 2 of [B] shows that one layer of the ImageNet classification pre-trained backbone can activate appropriate regions of human faces, even though ImageNet does not have any face class nor a human class.
>
> The other main reason is the inherent task gap between the current SSL and 3DHPSE.
> A 3D human pose, one of the targets of 3DHPSE, is a complex set of multiple joints that relates to each other in the kinematic chain.
> However, the current methods of SSL have an instance-level learning characteristic, which is unaware of joints that add up to a human pose.
> They augment an instance (human)-centered image and try to encode a common feature from the augmented and unaugmented images.
> Their mechanism could make an image backbone learn visual representations that could generalize well to instance-level downstream tasks, such as classification and object detection.
> However, the visual representations essentially lack understanding of the fine-level semantic information, the human joints.
>
> To empirically validate the two reasons discussed above, we designed a new pre-training approach that combines an SSL approach with 2D joint labels (2D pose).
> We call it _JointCon_.
> It applies contrastive learning to joint-level features, which are locally sampled image features using GT 2D joint locations following Pose2Pose (Moon et al., 2022a).
> The new approach shares the same network architecture, contrastive loss, and augmentations with PeCLR (Spurr et al., 2021).
>
> Two different versions of JointCon are experimented.
> The first version contrasts joint-level features of the same joint class from different images as other SSL methods (Chen et al., 2020b; Chen & He, 2020; Chen et al., 2020c; He ́naff et al., 202, Spurr et al., 2021).
> The 'positive' samples of an anchor joint-level feature are only from augmentations, and joint-level features of the different joint classes and the same joint class from different images are treated as 'negative' samples.
> The second version treats joint-level features of the same joint class from different images as 'positive samples', inspired by the supervised contrastive learning [A].
> The first version of JointCon achieves 56.71mm/59.59mm PA-MPJPE on 3DPW, 47.99mm/56.55mm PA-MPJPE on Human36M, and 67.85/67.38 3DPCK on MuPoTS, when pre-trained on MSCOCO and fine-tuned on 100\%/10\% data (Human36M + MSCOCO).
> The second version of JointCon achieves 54.25mm/57.19mm PA-MPJPE on 3DPW, 45.52mm/54.97mm PA-MPJPE on Human36M, and 68.87/67.98 3DPCK on MuPoTS, when pre-trained on MSCOCO and fine-tuned on 100\%/10\% data (Human36M + MSCOCO).
> The JointCon's second version outperforms the first version in all three benchmarks.
>
> [A] Khosla et. al. Supervised contrastive learning. In NeurIPS, 2020.
> [B] Yosinski et. al. Understanding neural networks through deep visualization. In ICML workshop, 2015.

---

> ### Author Response · Authors · 2022-11-17
> **Follow-up**
>
> Dear reviewer,
>
> We would like to follow up to check if your concerns have been addressed. In the previous response, we made the following updates/clarification:
>
> - Regarding your concern about the reasoning for the poor performance of SLL pre-training, we designed new experiments in **s7cS-1** that scrupulously analyze the limitations of the current SSL for 3DHPSE. The experimental results and analysis are in line with those in Sections 4.2 and 4.3 and will be included in the final version of this paper.
> - We answered the confusion about the semi-supervised setting in **s7cS-2**.
>
> We are happy to answer further questions.

---

### Author Response · Authors · 2022-11-13
**Overall Response to Reviewers**

We thank the reviewers for their insightful comments and remarks.
All reviewers agree that the paper presents a well-organized and comprehensive (EdAX) study on the effectiveness of the pre-training for the 3DHPSE task, which is new (C9uS), and that it will provide useful guidance to the community (jtxR), as well as the valuable analysis and discussions.
Nevertheless, a few concerns were raised about the current SSL's negative results and the novelty of this paper.
We address them in the following with a more in-depth investigation on the SSL and a clear-cut explanation of the difference with the previous literature.

---

### Decision · Program_Chairs · 2023-01-20

**Decision:**

Accept: poster

**Justification For Why Not Higher Score:**

The paper provides extensive experiments and achieves some insightful conclusions. It is interesting and beneficial for the community, but at the same time there remains concerns on the significance of the result.

**Justification For Why Not Lower Score:**

N/A

**Metareview: Summary, Strengths And Weaknesses:**

This paper conducts an empirical study on how to pre-train visual representation for better human pose and mesh estimation. All the reviewers agree that the paper provides thorough experiments and the findings are interesting and insightful. All the reviewers agree to accept the paper and the AC agrees.

**Note From Pc:**

if the above contains the word "oral" or "spotlight" please see: "oral" presentation means -> notable-top-5% and "spotlight" means -> notable-top-25%. As stated in our emails, we are disassociating presentation type from AC recommendations